# Aerobic Biostabilization of the Organic Fraction of Municipal Solid Waste—Monitoring Hot and Cold Spots in the Reactor as a Novel Tool for Process Optimization

**DOI:** 10.3390/ma15093300

**Published:** 2022-05-04

**Authors:** Sylwia Stegenta-Dąbrowska, Peter F. Randerson, Andrzej Białowiec

**Affiliations:** 1Department of Applied Bioeconomy, Wrocław University of Environmental and Life Sciences, 37a, Chełmońskiego Str., 51-630 Wrocław, Poland; sylwia.stegenta@upwr.edu.pl; 2School of Biosciences, Cardiff University, Sir Martin Evans Building, Museum Avenue, Cardiff CF10 3AX, UK; randerson@cardiff.ac.uk

**Keywords:** air flow rate, composting, municipal solid waste, monitoring, optimization, spatial distributions

## Abstract

The process of aerobic biostabilization (AB) has been adopted for treatment of the organic fraction of municipal solid waste (OFMSW). However, thermal gradients and some side effects in the bioreactors present difficulties in optimization of AB. Forced aeration is more effective than natural ventilation of waste piles, but “hot and cold spots” exist due to inhomogeneous distribution of air and heat. This study identified the occurrence of hot and cold spots during the OFMSW biostabilization process at full technical scale. It was shown that the number of hot and cold spots depended on the size of the pile and aeration rate. When the mass of stabilized waste was significantly lower and the aeration rate was two-fold higher the number of anaerobic hot spots decreased, while cold spots increased. In addition, the results indicated that pile construction with sidewalls decreased the number of hot spots. However, channelizing the airflow under similar conditions increased the number of cold spots. Knowledge of the spatial and temporal distribution of process gases can enable optimization and adoption of the OFMSW flow aeration regime. Temperature monitoring within the waste pile enables the operator to eliminate undesirable “hot spots” by modifying the aeration regime and hence improve the overall treatment efficiency.

## 1. Introduction

One of the crucial strategies for waste management both in Poland and around the world is the treatment of the organic fraction of municipal solid waste (OFMSW). Basic assumptions and objectives in the area of municipal solid waste (MSW) management, adopted both in the EU [1,2,3] and in national legal acts, aim to limit the use of landfilling as a means of reducing the organic fraction [4]. One of the solutions is the mechanical and biological treatment (MBT) of mixed municipal waste [5]. In Poland, at the end of 2016, there were 192 MBT plants with a total mechanical capacity of around 11 million tons of waste per year [6]. Mostly, the biological process of aerobic biostabilization (AB) has been adopted for OFMSW treatment [7]. However, optimization AB of OFMSW is difficult, due to its heterogeneity, thermal gradients, and some side effects in the bioreactors. Additionally, bioreactors treating many tons of OFMSW are poorly equipped with sensors for temperature, oxygen, or moisture, so that the plant operator has little control over most of the waste mass.

Gas flow patterns within the waste have a large influence on heat and mass transfers. Consequently, O_2_ supply, moisture and temperature distribution have a large impact on the end-product quality (kinetics of biodegradation; stage of stabilization; hygienization of the compost), as well as on the environmental impact of the treatment (gaseous emissions and odors) [8]. Within the AB process, gases in the pores are heated due to microbial activity [9], which reduces air density inside the pile pores and increases partial pressure, creating a flux from the bottom to the top layers in naturally aerated piles [10]. However, the gas production rate differs with spatial gradients because of ineffective mixing (inhomogeneity) and compaction effects, resulting in concentration gradients which drive gas diffusion and transfer inside the pores [11]. The effect can be visible even in well-mixed waste, especially in municipal solids which are characterized by huge inhomogeneity. Anaerobic areas (so-called “hot spots”) need to be diagnosed and eliminated because gaseous biostabilization products (CO, CO_2_, CH_4_) are substantial threats to both low treatment efficiency and human and environmental protection.

The technique of forced aeration has been adopted in MSW treatment, as it is more effective than natural (passive) ventilation for the aerobic metabolism of microorganisms, removal of water, and control of the temperature of the system [12]. It has been reported that the O_2_ content in the air space has no significant effect on the biological degradation efficiency until it falls below 5% in the composting matrix [13]. During the aeration process O_2_ content may rise above 15%, gradually decreasing again after the air blower stops, although O_2_ content may support aerobic bioactivity for an extended period. Furthermore, it has been demonstrated that intermittent aeration could reduce NH_3_ loss from the composting system compared to continuous aeration [14]. Our previous research showed that low O_2_ concentration could also favor CO production [15,16]. Gaseous emissions during biostabilization not only reduce the compost quality, but also cause atmospheric pollution [17]. Therefore, an improved AB process is urgently needed.

Even if forced aeration is more effective than natural ventilation, “hot spots” in OFMSW exist due to its heterogeneity [15], as well as aeration rate and reactor design. The proper design and operation of a biostabilization project requires an understanding of the dynamics of biostabilization [18]. In particular, the process depends on the abundance and activity of microorganisms, which are mainly affected by temperature, moisture, readily degradable organic content, O_2_ level and its diffusion in the matrix, and presence of inhibiting compounds. Without frequent turning in a static composting system, or in the absence of dynamic aeration, significant spatial differences in these parameters resulting from one-directional air flow will impact the spatio-temporal dynamics and hence the uniformity of the compost product [16,19]. An appropriate reactor, adapted to the characteristics of the waste, should not only maintain appropriate levels of O_2_ and temperature, but also allow for uniform distribution within the pile [20]. Hence, monitoring the spatial and temporal distribution of pore gas concentrations is an important method for evaluating and optimizing the aeration strategy and reactor design and operation [21].

Adoption of a flow aeration regime, together with knowledge of the spatial and temporal distribution of process gases, and temperature can enable optimization and control of parameters such as temperature, O_2_ or even CO. Monitoring the anaerobic “hot spots” during AB of OFMSW may be a useful tool in mitigating emissions of gaseous pollutants and optimizing the biostabilization processes.

The aim of this study was to investigate the spatial and temporal distribution of temperature and pore gas (O_2_, CO_2_, and CO) concentrations in relation to anaerobic “hot spots”. Spatial and temporal variability of gas concentrations and temperatures were determined at full technical scale in a municipal biostabilization plant.

## 2. Materials and Methods

### 2.1. Characteristics of Organic Fraction of Municipal Solid Waste

The material used in the research was the Organic Fraction (undersize fraction < 80 mm) of Municipal Solid Waste (OFMSW) originated from Warsaw, Poland. The waste was characterized by: moisture content in accordance with Polish standard PN-EN 14346:2011, volatile solids (VS) in accordance with PN-EN 15169:2011, the Total Organic Carbon (TOC) in accordance with Polish standard PN-EN 15936:2013, pH in accordance with Polish standard PN-EN-15011-3:2001, and morphological composition in accordance with Polish standard PN-93/Z-15006. The tests were performed each time before (6 samples) and after the biostabilization process was completed (6 samples). This gave 12 samples in total.

Total mass of OFMSW loaded into the bioreactor was also recorded before and after the biostabilization process. Each OFMSW sample was collected according to the following procedure: from 4 places along the length of the reactor, in each place, ~10 samples were taken. All collected samples, from one reactor, at the same collection moment, were mixed and using the quartering method were assigned to a representative sample. Properties of the waste before and after treatment are summarized in Figure 1. Detailed analytical results are described in Section 3.1 and in Supplementary Tables S3 and S4 in our previous paper [22].

### 2.2. Process Configuration

Experimental monitoring of the AB process of OFMSW was performed between 24 April and 9 September at the Municipal Cleaning Company in Warsaw, Poland at full industrial scale. For each of the six static pile bioreactors, dynamic aeration was provided through three aeration channels (Table 1). The procedure for pile building was to collect waste with a loader (two–three days); cover the waste with a membrane; turn on blower to start aeration process. The membrane provides full protection against the weather, including precipitation. Bioreactor pile configuration differed in terms of initial mass (350–400 Mg in piles A1 and A2, and 600–700 Mg in piles B1, B2, C1, and C2) duration of the process (six weeks in piles A1, A2, and B1 or nine weeks in piles B2, C1, and C2) as well as reactor construction (the use of concrete sidewalls in piles C1 and C2) and the total amount of air forced into the pile (Figure 1). Weekly measurements of O_2_, CO_2_, and CO concentrations and temperature measurements inside the piles were performed. Outside temperature measurements were also taken.

The total experimental configurations are shown in Figure 1. Experimental details and the raw data obtained from the measurements can be found in [22].

### 2.3. Gas Concentrations and Temperature Measurements

Measurements were taken along the length of each reactor at four locations: 2.5 m, 17.5 m, 32.5 m, and 47.5 m from the fan (Figure 2A). At each location (cross-section of the pile), measurements were made at three heights on both sides of the pile (Figure 2B). Additionally, on one side of pile one “deep measurement” in each cross-section was made. The height and depth of sampling were determined individually for each bioreactor according to their dimensions, as shown in Table 2. Measurements of gas concentration and temperature were made with a 3.5 m long steel probe (lance), perforated at the end. As shown in Figure 2, in piles A1, A2, B1, and B2 (Figure 2B), the gas/temperature sampling probe was inserted parallel to the ground, whereas in piles C1 and C2 (Figure 2C) the lowest measurement (H1) was made at an angle of 45° to avoid the sidewall. Each measurement was done outside the insulating layer of biostabilization waste, at least 1.25 m deep. In addition, a so-called deep measurement was taken on one side of the reactor at each distance point at the middle height (Figure 2B,C). The probe was connected to an electrochemical analyzer Kigaz 300 by Kimo (Kimo Instruments, Chevry-Cossigny, France) with a plastic hose as well as a thermocouple. The procedures for gas and temperature were as follows: analyzer was started; 2-min autocalibration; placing the probe in the measuring location; waiting for concentration values stabilization (typically within ~5 min). After each measurement, the probe was removed from the biostabilization waste for ~1 min, allowing the measured gas values to return to ambient air (gas concentrations equal to 0%/ppm). O_2_ and CO_2_ volumetric contents in piles were measured in % (±0.1%), but CO contents were measured in volumetric ppm ±1 ppm. The temperature inside the piles was measured with ± 1 °C precision. The total amount of sampling is shown in Table 3. Detailed measurements of gas concentrations and temperature in the reactors have been previously shown [22].

### 2.4. Spatial Distribution Modeling

The Surfer 10 program (Golden Software, version 8.0, Cracow, Poland) was used to visualize the raw data recorded in Supplementary Material Table S3 [22], for spatial and temporal distributions of CO, O_2_, and CO_2_ gases and temperature in the piles in each cross-section and each measurement time, using a color scale.

Gas compositions and temperature values at each sample location within the piles (Figure 2 and Table 2) enabled spatial distribution modeling of each parameter, using the Natural Neighbor mathematical function method for cross-sections and Radial Basic Function for longitudinal cross-sections. Boundary conditions external to the piles were defined by the composition of atmospheric air for gases and external temperature, as well as corresponding measurements in the aeration channels below the piles (Figure 2B,C). A total of 172 Figures were generated, to represent the spatial distribution models Figures 5–8 (Appendix A: 1012 drawings, including 668 cross-sections and 344 longitudinal sections). Each figure presented 4 cross-sections: (a) 2.5 m, (b) 17.5 m, (c) 32.5 m, (d) 47.5 m; and 2 longitudinal sections: (e) left, (f) right. As proposed in the previous article [22], incomplete and uncertain data were rejected in the modeling process, hence the total number of Figures generated is fewer than the collected data (Table 3).

## 3. Results and Discussion

### 3.1. Waste Properties

As shown by other authors, the composition of the waste used for composting and its mixing has a great impact on the chemical and microbiological changes during the process [23,24]. In all OFMSW reactor piles, moisture contents (36–44%; Figure 1), were below the optimal range of 50–60%, according to Liang et al. (2003), but removal of moisture was more than two-times higher in piles A1 and A2, with a lower initial mass and a high rate of airflow > 12,000 m^3^∙Mg^−1^ (Figure 1) [25]. In reactors B1, B2, C1, and C2, the reduced airflow (Figure 1), and the relatively comparable ambient temperature (Figure 3) greatly reduced the loss of moisture in the stabilized material (23–31% of initial value compared to 61–71% in piles A1/A2). Greater removal of moisture in a smaller composting pile was also obtained by Ermolaev et al. (2012) [26]. However, this was much higher than the small decrease (~10%) obtained by Mulbry and Ahn, (2014), using much smaller piles (volume ~1.9 m^3^) [27]. The effectiveness of the process of moisture removal apparently depends on the amount of air blown in relation to the total waste mass of the pile.

Reduction in total mass of the OFMSW was greater in piles A1, A2, and B1 (34–36%), where the process time was longer (9 weeks), compared to the other piles (6 weeks). The final content of organic material (VS: 32–38% d.m.; TOC: 18–22% d.m. (Figure 1) was about half that recorded by [28] during MBT composting of municipal solid waste, and similar values were observed in Komilis et al. (2012) in a larger-scale installation (capacity 250,000 Mg∙year^−1^) [29]. The content of organic substances (both VS and TOC) is typical for MBT composting plants [30]. As with moisture removal, organic matter was most effectively reduced in piles A1/A2 with relatively small size and high airflow. Similarly, Hu et al. (2003) concluded that, for the decomposition of TOC, the moisture and fraction size have a greater influence than the process temperature [31].

The pH value of initial samples of OFMSW was very similar in all analyzed piles (5.4–5.6) (Supplementary Material Table S3 [22]), whereas at the end of the process the pH increased towards neutrality. These values and their change during the biostabilization process are typical for MSW [32].

The fine fraction < 20 mm comprises a large proportion (66–77%) of the OFMSW (Figure 4), with variable amounts, typical for municipal waste, of kitchen waste, paper and plastic and small proportions of textiles (0.3–2.2%), glass (3.8–5.3%), metals (0.8–2.9%), other organic (0.9–2.4%), other minerals (1.7–3.8%), and other materials (0.3–2.6%). After the process, a decrease in the content of kitchen waste and paper was observed in A1, A2, and B1, whereas in B2, C1, and C2, the content remained unchanged or increased, which suggests that the degradation process is more effective with a longer operation time.

The < 20 mm fraction in the biostabilized material was at least 65% (Figure 4), higher than values of 50%, typical of Polish conditions [33]. In other studies, the content of organic waste was 60–67%, paper 15–17%, glass 7–8%, plastic 6–7%, and metal 3–5% [34]. The efficiency of waste mass removal, ranging from about 20 to 30% (Figure 1), was close to the average values obtained in other similar installations, operating biostabilization technology for municipal solid waste in Poland [35].

### 3.2. Spatial and Temporal Distribution Changes

Appendix A show the spatial and temporal changes in piles as follows: temperature (Appendix A) and the concentration of O_2_ (Appendix A), CO_2_ (Appendix A) and CO (Appendix A) in individual sections, at weekly intervals during the biostabilization process in individual piles. The right side of the pile was facing to the south and after the semi-permeable membrane was removed, it was exposed to direct sunlight on the day of the measurement, but measurements were taken inside the pile. No heating effect of exposure to the sun was observed inside the pile.

As the terms “hot” and “cold” spots in waste reactors were not found in the literature, they are defined here as localized areas within the pile where all three measured parameters (Table 4) were met in the same place and at the same time. All hot and cold spots recorded in this study are shown in Table 5 (cold spots) and Table 6 (hot spots). Locating these points is of great practical importance.
Hot spots—areas of elevated temperature, liable to loss of stability of the biocenosis (high temperature > 60 °C with low oxygen < 15% can cause the loss of valuable microorganisms).Cold spots—areas where conditions for sanitizing the waste by inactivating potential pathogens and parasites cannot be maintained (low temperature < 30 °C reduce the efficiency of microorganisms or induce the switch to spore formation).

In either case, the quality of stabilized waste may be reduced, or the process time increased.

#### 3.2.1. Spatial and Temporal Distribution of Temperature

The pattern of temperature change during the waste composting process was expected to follow the “classic” pattern described by Cooperband, (2002) [36] and Kowal et al. (2017) [37], with an initial rapid increase in temperature (1st phase of composting) followed by a gradual drop in temperature during maturation/cooling (2nd phase of composting). In contrast, temperatures in piles A1 and A2 (Appendix A) remained close to ambient (Figure 3) throughout the entire prism profile (~20 °C) during the first two weeks of the process. This lag phase resulted from the high aeration rate (>12,000 m^3^∙Mg^−1^) and the small mass weight in the reactors (300–400 Mg: ~40% less than normal for this treatment plant). From day 20, in both piles, internal temperatures increased to ~60–70 °C (Appendix A), three hot spots where found (high temperature coupled with low O_2_: Table 5), and further small increases were observed from weeks 5 to 7. Similar temperature patterns were observed in a reactor without forced aeration by Jiang et al. (2015) [38], during composting of green waste [39], kitchen waste [40], and vermicomposting of duck manure [41]. However, Mulbry and Ahn (2014) showed that, in much larger scale piles, passive aeration allows them to heat up as quickly as during forced aeration [27].

In the other piles, an increase to 60 °C was observed at the beginning of the 2nd week. This high temperature was maintained in the piles until the end of the process ~day 50 for piles A1, A2, and B1 (Appendix A), ~day 35–40 for B2 and C1 (Appendix A), and ~day 30 for C2 (Appendix A). Temperatures in piles B1, B2, C1, and C2 followed the expected 2-phase pattern.

An interesting trend was observed in pile A1—the highest temperatures, even >70 °C, were found at the end furthest from the fan (Appendix A) from where the heat spread towards the front of the pile. This could be due to a poorer air supply to the far end of the pile, resulting in self-heating and poor heat removal. In pile A2 this did not occur (Appendix A), despite a similar air load and a mass of waste greater by about 15% greater. In addition, lower aeration values were noted by about 1.5 m^3^∙Mg^−1^∙h^−1^ in the first 3 weeks of the research (Supplementary Material Table S1 Stegenta-Dąbrowska et al. (2020)) [22]. The opposite effect was noticed in pile B2 (Appendix A) which showed greater cooling close to the fan on the final day. This may be the result of faster decomposition of waste located near the fan, which at the end of the process significantly reduced endothermic processes due to the lower activity of microorganisms.

At the same time a few cold spots were observed (Table 6), located mainly at the bottom of pile, due to the forced aeration. The cooling effect of aeration by the three channels below each pile is clear in the visualization (Appendix A). Typically, this occurs in both the first week of biostabilization when internal heat takes longer to accumulate, and in the final week when the need for O_2_ is lower.

The observation that, in piles A1 and A2 high temperatures > 50 °C remained during the final days of the process despite the continuous high air stream (Appendix A) indicates high bio-activity of the stabilized waste and consequent high O_2_ demand. However, it could result from the unexpected initial two week lag phase in heating the pile.

The highest observed temperatures occurred in pile B1 (from day 16: >70 °C inside the pile, >60 °C at the edge), which contained the highest TOC and organic substances (VS) of all analyzed samples, as well as having twice the mass of piles A1 and A2, but half the rate of aeration.

In piles C1 and C2 the sidewalls apparently contributed to an overall reduction of temperature (Appendix A). The cooling of the waste at the border with the walls is visible, especially at the beginning and at the end of the process—a few cold spots were identified near to the sidewalls and at the bottom (Table 6). Despite favorable conditions (large mass of waste; less aeration), no elevation of temperature was noticed.

Experiments carried out under similar conditions by Ermolaev et al. (2012) showed that, despite continuous operation, fans may be unable to maintain an appropriate temperature [26]. Our observations of cold spots (low temperatures) at the base of the pile (e.g., Figure 5; Appendix A and Table 6) indicates the impact of pumping large amounts of air, resulting in removal of warm air from the center and overall cooling of the pile. Similar effects of lowering temperature by increased aeration were observed by Shen et al. (2011) and Sołowiej et al. (2010) [42,43]. In bigger piles, B1–2 and C1–2, the optimal temperature for the biostabilization process, around 60 °C, was reached much faster, an effect also noted by [44]. Although heat contributes to elimination of pathogenic organisms (Stentiford, (1996) it does not ensure maximum mass removal [45]. According to many authors, greater mass reduction can be achieved at temperatures between 40 and 60 °C [45,46,47]. On the other hand, Richard, (1993) claims that maintaining the temperature in the range of 56–70 °C for too long (over a week) reduces biodiversity and increases the intensity of odor compounds [47].

**Figure 5 materials-15-03300-f005:**
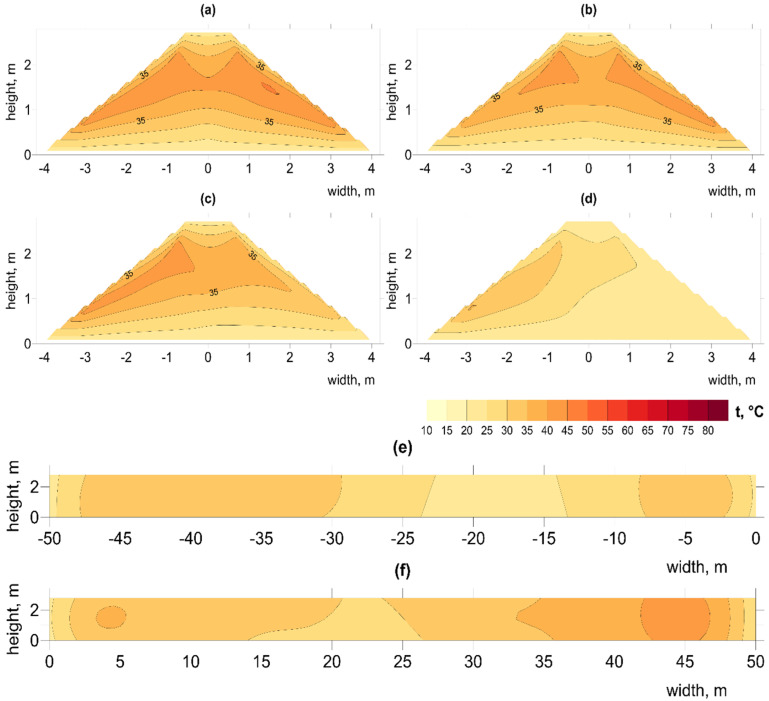
Spatial distribution of temperature changes on day 1 in pile B1, at distances from the aeration fan (**a**) 2.5 m, (**b**) 17.5 m, (**c**) 31.5 m, (**d**) 47.5 m, longitudinal sections (**e**) left (**f**) right. Illustration of the problem with irregular temperature in pile.

#### 3.2.2. Spatial Distribution of O_2_ and CO_2_ Concentration

Despite the high aeration in piles A1 and A2 during the first two weeks, low O_2_ concentrations were recorded, together with low temperatures (Appendix A). The high rate of airflow would have created optimal conditions for the activity of psychro- and mesophilic microorganisms at that time, leading to depletion of oxygen by respiration. However, the inhomogeneity of the OFMSW and differences in bulk density could also affect airflow through the pores creating local “hot spots”. O_2_ levels were particularly low in most cases at the center of the pile early in the process (e.g., Appendix A), but hot spots were discovered in pile A2 at 2.5 m length on day 25 (Table 5). In pile A1, reduced O_2_ concentrations (~15% lower) occurred on the left side of the pile (Appendix A), corresponding to the higher temperatures in the same areas (Appendix A). The above examples were all identified as a hot spot (Table 5). Low O_2_ concentrations result from the high intensity of biodegradation early in the process (Jiang et al. (2015) especially if aeration is insufficient [38]. Mohajer et al. (2010) note that the O_2_ consumption strongly increases in the first 4 days of composting to as much as 40 mmol∙h^−1^∙kg^−1^ d.m., and then decreases with the duration of the process [48].

O_2_ concentrations increased during the process, related to decreasing O_2_ consumption, while supply of forced air remained stable (Supplementary Material Table S3, [22]). From day 30, O_2_ concentrations > 17% were observed in most piles, indicating excellent oxygenation of the waste and sufficient air forced into the pile to promote aerobic digestion.

Data obtained in this experiment shows a generally better aeration system (few locations with less than 15% O_2_) compared to composting in active aerated and static piles reported by Szanto et al. (2007) [49]. Although the total aeration intensity per waste mass m^3^·Mg^−1^ (Supplementary Material Table S1, [22]) was below the recommended level, >10 m^3^∙Mg^−1^∙h^−1^ [4], this did not appear to affect the O_2_ concentration observed in the piles.

Extremely low O_2_ concentrations < 5% were observed only at the beginning of the process in pile B1 (Appendix A). Low concentrations (<10% O_2_) also occurred in the center of this pile up to day 40 (Figure 6 and Figure 7; Appendix A) coinciding with high temperatures (Figure 5; Appendix A). Compared to other piles, B1 contained the greatest number of such hot spots (Table 5). This may indicate that, with an increased amount of waste and high TOC content, the air flow was insufficient during the most intense phase, up to the 4th week of the decomposition process [50].

In pile B2, lower concentrations of O_2_ were observed at its sides (Appendix A) with hot spots mainly in the center of the pile (Table 5), which may also confirm insufficient air supply where the total waste mass exceeds 600 Mg. The influence of reactor design during biological waste treatment process has been noted by Mason and Milke, (2005) [20]. Another explanation could be the structure of the waste, consisting mainly of waste fractions < 20 mm (Appendix A), which could reduce the free air spaces and obstruct the air from the aeration channels. Whether aeration is passive or active, airspace within the substrate plays an important role in the composting process [51]. Air porosity influences not only air permeability, but also determines O_2_ transport and the removal of water and heat from the pile.

Very high concentrations of O_2_ > 18% were observed in C1 and C2, where sidewalls were constructed. Low concentrations of O_2_ < c8% (Appendix A) and high concentrations of CO_2_ (Appendix A) occurred at only a few points (e.g., the cross-section of 17.5 m) indicating hot spots (Table 5). CO_2_ concentrations are consistent with results obtained by Clemens and Cuhls, (2003) from various types of piles composting municipal solid waste [52]. The spatial distribution of CO_2_ showed an inverse relationship with O_2_ (e.g., in pile A1, higher CO_2_ concentrations occurred on the left side of the pile together with high temperature and lower O_2_), which is typical for aerobic waste treatment [53,54]. The highest content of CO_2_ > 10% was observed in B1, mainly up to week 2, but levels were mostly very low (2 to 3%) and occurred in the center of the pile as single hot spots. Similarly, the highest CO_2_ concentrations were observed in the first phase of composting a mixture of manure and sawdust, and then its gradual reduction as the compost matures [55], and at a small scale during home composting [56].

The influence of the sidewalls in piles C1 and C2 was also noticed as an increase in the occurrence of hypoxic zones near to the border of the walls. This could have resulted from poorly located aeration channels, which were originally designed for non-compacted piles, or from the small amount of air supplied to the pile. Despite such issues of reactor design, an active aeration system is essential, since CO_2_ may increase to over 25% with inadequate aeration [49].

**Table 5 materials-15-03300-t005:** Localization of hot spots during biostabilization process.

Hot Spots				
Pile	Place (Length), m	Time, Day	Localization	Appendix A
A1	2.5; 17.5, 32.5	20	In the center	Appendix A
A1	2.5, 32.5	28	In the center	Appendix A
A1	32.5	41	In the top left corner	Appendix A
A2	2.5	25	In the center of pile	Appendix A
A2	2.5	32	In the top left corner	Appendix A
A2	17.5	47	In the top left corner	Appendix A
B1	17.5	9	In the center of pile	Appendix A
B1	2.5, 17.5, 32.5, 47.5	16	In the center and left side	Appendix A
B1	2.5, 17.5, 32.5, 47.5	22	In the center and left side	Appendix A
B1	2.5, 17.5, 32.5, 47.5	31	In the center and left side	Appendix A
B1	2.5, 17.5, 32.5, 47.5	41	In the center and left side	Appendix A
B1	17.5	57	In the left bottom corner	Appendix A
B2	2.5, 32.5, 47.5	8	In the center and left side	Appendix A
B2	32.5	16	In the left bottom corner	Appendix A
B2	17.5, 32.5, 47.5	24	In the center and right side	Appendix A
B2	32.5	43	Right down corner	Appendix A
C1	32.5	28	In the center	Appendix A
C2	17.5	9	On the right side	Appendix A
C2	32.5	21	On the left side	Appendix A
C2	17.5	35	On the left and right corner	Appendix A

**Table 6 materials-15-03300-t006:** Localization of cold spots during biostabilization process (excluding data from first three weeks of biostabilization in piles A1 and A2, due to low temperature in all piles).

Hot Spots				
Pile	Place (Length), m	Time, day	Localization	Appendix A
A1	2.5; 17.5, 32.5	20	In the left side in the bottom	Appendix A
A1	2.5, 32.5	28	In the bottom	Appendix A
A2	2.5, 17.5	17	In the bottom	Appendix A
A2	2.5, 17.5, 32.5	25	In the bottom	Appendix A
A2	2.5, 17.5	38	In the bottom	Appendix A
A2	2.5, 17.5, 32.5, 47.5	57	In the bottom	Appendix A
B1	32.5, 47.5	1	In the bottom and in the center	Appendix A
B2	2.5, 17.5, 32.5, 47.5	1	In the bottom	Appendix A
B2	2.5	43	In the bottom and in the center	Appendix A
C1	2.5, 17.5	28	left side, near to sidewalls and in the bottom	Appendix A
C1	2.5, 32.5, 47.5	35	left side, near to sidewalls	Appendix A
C1	2.5, 32.5, 47.5	42	Left and right side, near to sidewalls	Appendix A
C2	2.5, 17.5, 32.5, 47.5	1	In the bottom	Appendix A
C2	32.5, 47.5	28	In the bottom	Appendix A
C2	2.5, 32.5, 47.5	35	In the bottom, left side, near to sidewalls	Appendix A
C2	2.5, 32.5, 47.5	42	In the bottom, left side, near to sidewalls	Appendix A

It has been shown that the number of hot and cold spots depended on the size of the pile and aeration rate. Comparison between piles A1–2 and B1–2 shows that when the mass of stabilized waste is significantly lower and the aeration rate is two-fold higher the number of hot spots decreases, while cold spots increase. In the case of piles A1–2, the number of hot spots was 6 and 3, respectively, while in piles B1–2, 16 and 8. The opposite situation was in the case of cold spots 5–11 (A1–2) and 2–5 (B1–2) (Table 5). It shows that the application of hot and cold spot monitoring may be a useful tool for the optimization of the AB process. The elimination of hot and cold spots should be the aim, to achieve proper conditions for an efficient process, however, it requires further investigation. Additionally, our results indicated that construction with sidewalls (piles C1–2) decreased the number of hot spots to just 1 and 3, respectively (Table 4). This decrease may be due to improved airflow through the waste by eliminating air escape near the base of the pile. However, in piles B1–2, with similar waste mass and airflow rate, the channelized airflow increased the number of cold spots to 8 and 12, respectively (Table 5), indicating that differing reactor constructions also requires an optimal airflow rate to avoid inadequate aeration.

**Figure 6 materials-15-03300-f006:**
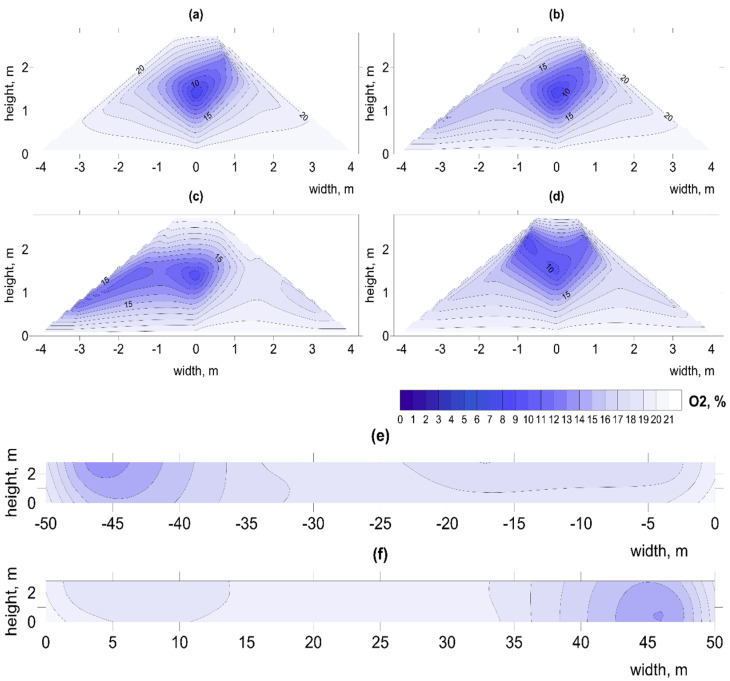
Spatial distribution of O_2_ changes on day 22 in pile B1, at distances from aeration fan (**a**) 2.5 m, (**b**) 17.5 m, (**c**) 31.5 m, (**d**) 47.5 m, longitudinal sections (**e**) left (**f**) right. Illustration of low O_2_ concentration in center of pile.

**Figure 7 materials-15-03300-f007:**
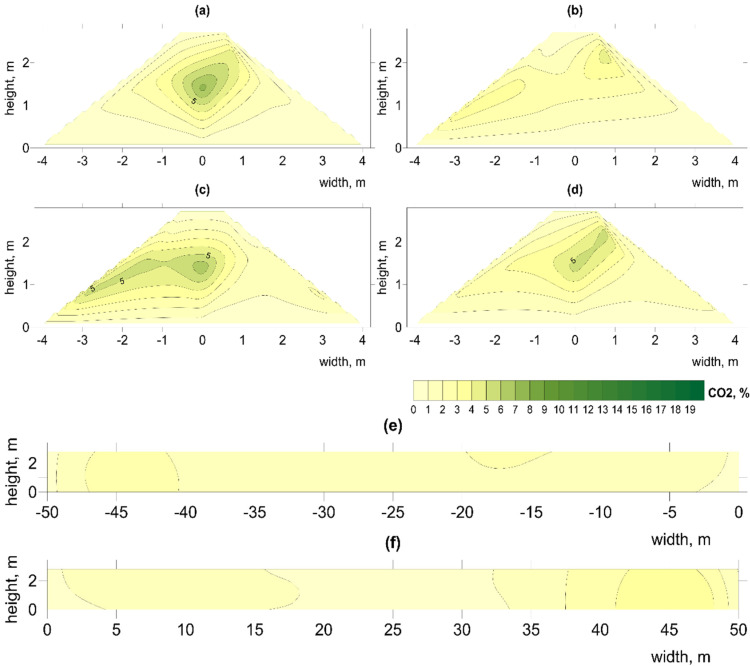
Spatial distribution of CO_2_ changes on day 22 in pile B1, at distances from aeration fan (**a**) 2.5 m, (**b**) 17.5 m, (**c**) 31.5 m, (**d**) 47.5 m, longitudinal sections (**e**) left (**f**) right. Illustration of high CO_2_ concentration in center of pile.

#### 3.2.3. Spatial Distribution of CO Concentration

CO was widely distributed throughout the entire process in all piles. In most cases the highest concentrations were observed in the center of the piles and during the first two weeks, in accordance with research carried out on other materials such as organic waste [57], green waste with sewage sludge [16,58], and municipal waste [15]. Similar relationships were noted by Boldrin et al. (2009) [59] during the composting of green waste, and Andersen et al. (2010) [39] who observed increased CO concentration in the early stages, which continued in composted material for a year.

Low concentrations, up to about 200 ppm, were observed during day 1 of the process in piles A1, A2, B1, and C1 (Appendix A), while in B2 and C2 there were unusually high values from the start, even over 1000 ppm, in the whole volume of the pile (Appendix A). High concentrations of CO throughout the pile during the whole process indicate the dynamic nature of CO production, resulting from the relatively large inhomogeneity of the waste material used.

High CO concentrations were clearly associated with locations of highest temperature. In the case of pile A1, following the initial lag phase, significant increases in both CO and temperature occurred from day 20 of the process (Appendix A). The highest concentration of CO was observed first at the beginning of the process (1st week, near the fan), which could be associated with the low O_2_ content, and at the far end of the pile over the whole duration. Around week 3, increased CO concentration was also noticed. In piles B1 and C1, from the first days of the process, temperatures > 60 °C were associated with very high concentrations of CO (>1000 ppm) (Figure 8; Appendix A). Similarly, Phillip et al. (2011) observed high CO concentrations during the first 6 h [60]. Also, in pile B1, very low O_2_ concentrations (<5%) were observed in the early stages (Appendix A), while in pile C1, only single hot spots were observed near the sidewalls, together with a lower O_2_ concentration (Appendix A). Hellebrand, (1999) showed that intensive aeration stimulates microorganisms to produce CO, greatly increasing its concentration [61]. In the present and previous studies [16,57], CO production coincides with the highest temperatures in the piles (up to 1800 ppm at 75 °C), which implies a thermochemical basis for the process.

The zones of lower oxygenation and higher temperatures near the sidewalls of the reactor in piles C1 and C2 coincided with higher concentrations of CO (Appendix A). Research by Hellebrand and Kalk, (2001) linked the release of CO directly to the availability of O_2_ in the pile, indicating that CO production is favored under both hypoxic and high temperature conditions [62]. As shown in previous studies, CO is also formed under aerobic conditions, but biotic production is more favorable at lower temperatures <40 °C, whereas at >60 °C, CO production is more thermochemical [57]. This clearly indicates the need for a waste treatment technology that will not form hot spots. The use of technologies that homogenize municipal solid waste, before placing it in the reactor may prove effective [63].

In piles where sidewalls were not used (A1, A2, B1, and B2), the biggest concentrations of CO increased in the center of the piles (Figure 8; e.g., Appendix A) or at the top of the pile, but smaller concentrations were observed, mainly at the end of the process (Appendix A). This may result from the aeration channels, which promote faster decomposition inside the pile, followed by decomposition in the upper regions of the material.

Several studies conclude that, in the decomposition of organic materials, microbial activity has a significant influence on CO production [60,62,64,65]. High temperatures > 60 °C prevailing in most of the piles, combined with a pH of about 7 at the end of the process, were optimal conditions for colonization by CO-metabolizing microorganisms [66]. Such microorganisms could then reduce CO production after the 4th week of the process, despite the high temperatures still prevailing.

Research by Moxley and Smith, (1998) showed the importance of moisture content for CO concentrations from various types of soils, with an optimum of 15 to 25% moisture [67]. Above and below these levels, CO concentration was reduced. Piles B1, C1, and C2 recorded the highest concentrations of CO but the lowest moisture removal, suggesting that the optimal value of moisture (in relation to CO production) of the OFMSW is rather higher, between 30 and 35%.

## 4. Summary

Spatial analysis of concentrations of key gases within the mass of OFMSW enabled the efficient localization of all hot and cold spots in time and space, regardless of the experimental variant tested or the reactor construction. It was observed that the localization of hot and cold spots depends on biostabilization process parameters including, aeration rate, and mass of OFMSW, or the type of reactor modification applied. It was shown that to reduce the appearance of cold and hot spots, it is necessary to:-increase the mass of the stabilized waste as it provides greater stability of the process to external conditions;-increase aeration of waste to remove anaerobic zones.

In addition, the use of sidewalls in pile construction reduced the occurrence of hot spots, and may have the effect of increasing the frequency of cold spots near to walls.

It has been shown that simple research on spatial and temporal distribution of temperature and gas concentration during the OFMSW biostabilization process is advisable, especially in the case of introducing new systems for processing municipal waste. Performing the tests allows quick and easy localization of all hot and cold spots, discovery of possible design mistakes, and adjustment of the parameters of the biostabilization process to shorten it and optimize its final products.

The identification of “hot spots” requires action to eliminate them by modifying waste aeration or its mechanical turning. This is important, especially to eliminate harmful gases such as CO, which have been seen clearly in hot areas. It may indicate the domination of thermochemical processes over biological ones, as already observed in green waste. Less importance in the production of CO is ascribed to the concentration of O_2_ and CO_2_, which implies that temperature measurement, together with spatial simulation, may be more effective in finding process irregularities. Locating areas of increased temperature within the pile will enable their elimination and the reduction of harmful gases. The identification of hot and cold spots during AB of OFMSW may be a useful tool for process optimization and indication of problems related to reactor construction, which also opens a new approach for research.

## Figures and Tables

**Figure 1 materials-15-03300-f001:**
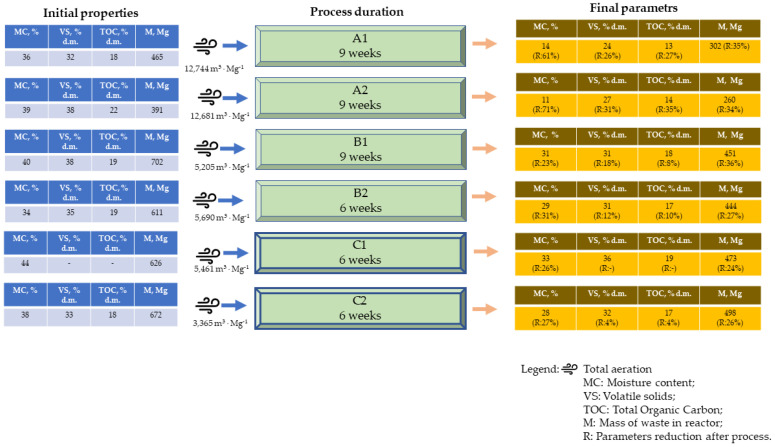
Process configuration and initial and final waste properties. Reactors A1, A2, B1, and B2 were covered by semipermeable membrane. Reactors C1 and C2 were constructed with concrete side walls and covered by semipermeable membrane.

**Figure 2 materials-15-03300-f002:**
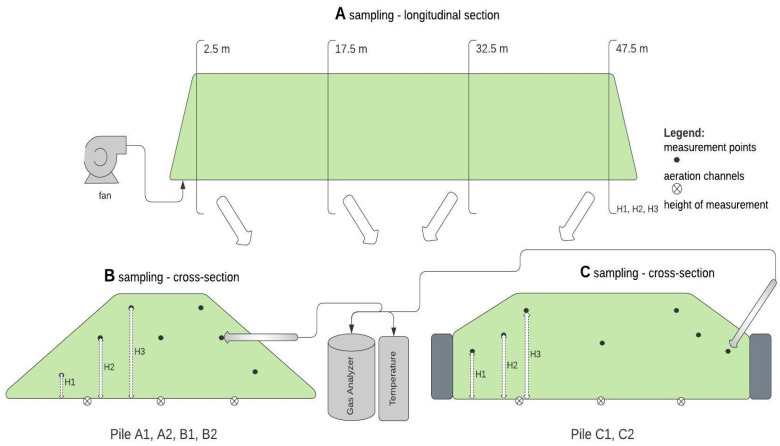
Process configuration and location of sampling points, (**A**) longitudinal repeats for each cross-section sampling point: (**B**) in piles A1, A2, B1, and B2; (**C**) in piles C1 and C2.

**Figure 3 materials-15-03300-f003:**
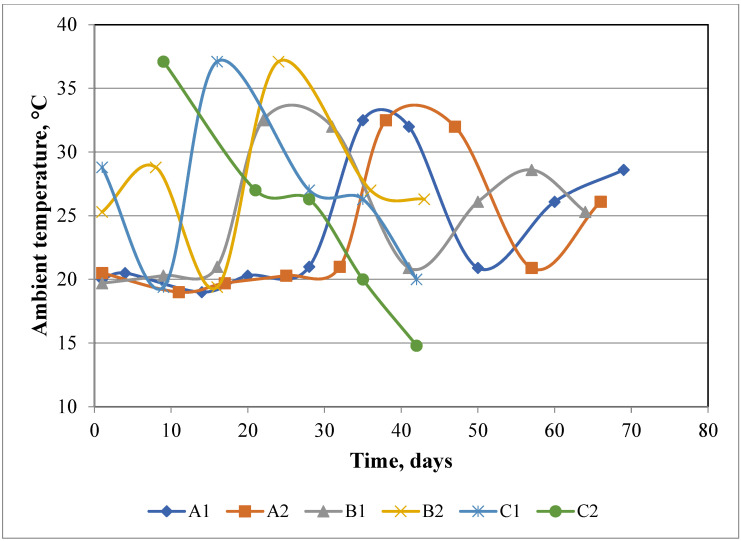
Ambient temperature around piles during the biostabilization process of Organic Fraction of Municipal Solid Waste (OFMSW).

**Figure 4 materials-15-03300-f004:**
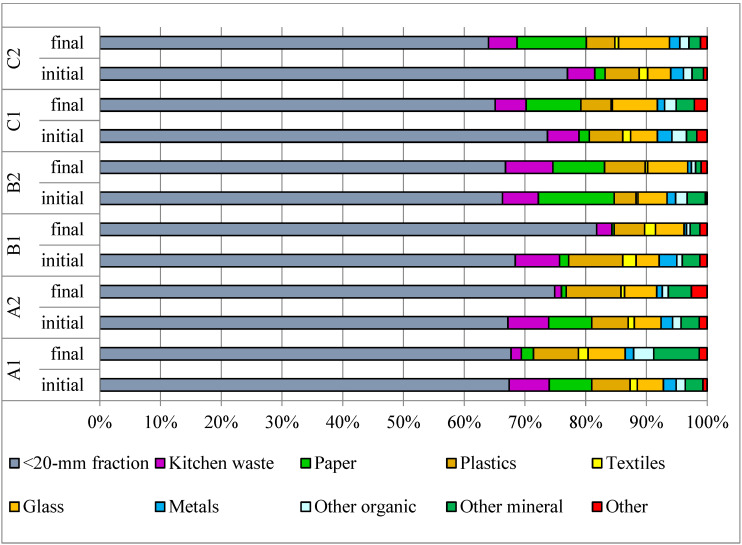
Changes in the morphological composition of OFMSW—samples were collected during the first and last day of the AB process and represent the entire reactor.

**Figure 8 materials-15-03300-f008:**
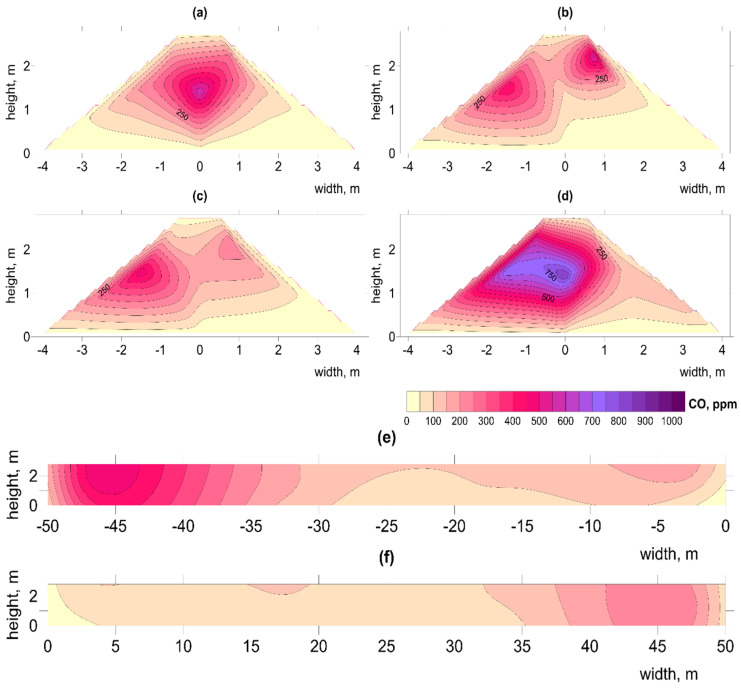
Spatial distribution of CO changes on day 16 in pile B1, at distances from aeration fan (**a**) 2.5 m, (**b**) 17.5 m, (**c**) 31.5 m, (**d**) 47.5 m, longitudinal sections (**e**) left (**f**) right. Illustration of high CO concentration in center of pile.

**Table 1 materials-15-03300-t001:** Technical specifications of aeration fan (blower).

Parameter	Value
Rotation, 1∙min^−1^	3480
Back pressure, Pa	3800
Rate of delivery, m^3^∙min^−1^	51
Power, kW	2.64

**Table 2 materials-15-03300-t002:** The dimensions of tested reactors [22].

Pile	Probing Height			Reactor Height, m	Reactor Width, m	Reactor Length, m
	H1, m	H2, m	H3, m			
A1	0.625	1.250	1.875	2.5	8.1	50.0
A2	0.625	1.250	1.800	2.1	8.0	50.0
B1	0.750	1.500	2.250	2.8	8.4	50.0
B2	0.750	1.500	2.250	2.9	9.0	50.0
C1	0.750	1.500	2.250	2.6	8.0	50.0
C2	0.750	1.500	2.250	2.5	8.0	50.0

**Table 3 materials-15-03300-t003:** Configurations of biostabilization piles, data acquisition cycles, and numbers of collected gaseous concentration samples and temperature measurements.

Pile	Process Start Date	Process Time, Days	Reactor Design	Number of Temperature and Gas Sampling Cycles	Number of Sampling Cross-Sections	Number of Sampling Pointsin Each Cross-Section	Number of Temperature and Gas Samples Collected
A1	24 April 2015	69	Membrane-covered pile	10	4	7	280
A2	27 April 2015	66	Membrane-covered pile	9	4	7	252
B1	13 May 2015	62	Membrane-covered pile	9	4	7	252
B2	15 July 2015	44	Membrane-covered pile	6	4	7	168
C1	22 July 2015	44	Membrane-covered pile with sidewalls	6	4	7	168
C2	30 July 2015	42	Membrane-covered pile with sidewalls	6	4	7	168
Total	-	327	-	46	24	42	1288

**Table 4 materials-15-03300-t004:** Characterization of hot and cold spots identified in this study.

Type of Spot	Parameter		
	Temperature, °C	O_2_, %	CO_2_, %
hot spot	>60	<15	>5
cold spot	<30	>15	<5

## Data Availability

Not applicable.

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
