# Peer review of "Aerobic Biostabilization of the Organic Fraction of Municipal Solid Waste—Monitoring Hot and Cold Spots in the Reactor as a Novel Tool for Process Optimization"

_materials, 2022, doi:10.3390/ma15093300_

Round 1

Reviewer 1 Report

1- What is the  knowledge gap in this work?

2-Why study the concentration of gases CO,CO2 and O2  only ?

3- Please add graphical abstract 

Reviewer 2 Report

Dear authors,

The manuscript you prepared is very interesting and very well prepared. However, I have few comments;

1-Have you thought of checking C:N ratios for both input and output materials? C:N ratio is one of the key indicator for compost stabilization.

2-I did not see any elemental analysis on the input material. It is always better to know the elemental analysis of input and out put products to see the values of C, N, K, TS, VS, TOC etc.

3-For your experimental work, it would have been very helpful if you had measured the porosity (or air porosity) of the compost material, which could give some hint for your hot and cold spots. 

I suggest minor revision before publication in Materials. 

Reviewer 3 Report

The study focuses on the hot and cold spots in aerobic biostabilization reactor, which is novel but not innovative. However,this study can provide a specific situation while biostabilization. Please read the paper again and have revisions, e.g. in line 142 'at least 1,25 m deep' may be corrected.   

Reviewer 4 Report

The composting process is a suitable technology for stabilizing organic matter from OFMSW. The paper deals with a relevant issue related to the difficulty of maintaining uniform conditions inside the reactor of composting. In this context, the detection of hot spots and cold spots is a pertinent issue.

 Minor aspects must be addressed:

-The composting process is in this paper called aerobic biostabilization (AB). However in the manuscript not clear that is the same process, because in some parts of the paper the reference is "composting" and in others AB. Some uniformization must be introduced in the paper.

-line 175, explain what is " longitudinal sections: e) left, f) right"

-line 186, Since many data are available, maybe Fig. 3 can be located in supplementary data, where the experimental points must not be joined by straight lines because between each point the temperature varied significantly.

-line 195, "final content of organic material (LOI: 32-38% d.m.;" Maybe VS (volatile solids) is a more appropriate designation than LOI.

- line 203, what is the meaning of "The pH reaction"?

- line 248, "3.2.1. Spatial and temporal distribution of temperature" Since several Figures are available in the supplementary material, related to temporal distribution, it may be interesting to show the variation of T on specific points of the piles over time. The idea is to illustrate the typical mesophilic- thermophilic-mesophilic profile of a mixture during the composting process.

-line 317, Why was selected "Spatial distribution of temperature changes on day 1 in pile B1" among many Figures?

-why CO2 profiles were not shown in the paper?

-line 403, Captions of Figure 6 and Figure 7 are incorrect.

-line 473, What is "...mass of stabilized waste"?

-line 475 "increase aeration of waste - to remove anaerobic zones." Why the concentration of CH4 was not measured in this work to detect anaerobic conditions?
